# Differential Detection of *Alternaria alternata* Haplotypes Isolated from *Carya illinoinensis* Using PCR-RFLP Analysis of *Alt a1* Gene Region

**DOI:** 10.3390/genes14051115

**Published:** 2023-05-20

**Authors:** Conrad Chibunna Achilonu, Marieka Gryzenhout, Gert Johannes Marais, Soumya Ghosh

**Affiliations:** 1Department of Plant Sciences, Division of Plant Pathology, Faculty of Natural and Agricultural Sciences, University of the Free State, Bloemfontein 9300, South Africa; 2Department of Genetics, Faculty of Natural and Agricultural Sciences, University of the Free State, Bloemfontein 9300, South Africa

**Keywords:** *Alternaria alternata*, black spot disease, *Carya illinoinensis*, PCR-RFLP, UPGMA analysis

## Abstract

Alternaria black spot disease on pecan is caused by the opportunistic pathogen *Alternaria alternata* and poses a serious threat to the local South African and global pecan industry. Several diagnostic molecular marker applications have been established and used in the screening of various fungal diseases worldwide. The present study investigated the potential for polymorphism within samples of *A. alternata* isolates obtained from eight different geographical locations in South Africa. Pecan (*Carya illinoinensis*) leaves, shoots, and nuts-in-shuck with Alternaria black spot disease were sampled, and 222 *A. alternata* isolates were retrieved. For rapid screening to identify Alternaria black spot pathogens, polymerase chain reaction–restriction fragment length polymorphism (PCR-RFLP) analysis of the Alternaria major allergen (*Alt a1*) gene region was used, followed by the digestion of the amplicons with *Hae*III and *Hinf*I endonucleases. The assay resulted in five (*Hae*III) and two (*Hinf*I) band patterns. Unique banding patterns from the two endonucleases showed the best profile and isolates were grouped into six clusters using a UPGMA (unweighted pair group method with arithmetic averages) distance matrix (Euclidean) dendrogram method on R-Studio. The analysis confirmed that the genetic diversity of *A. alternata* does not depend on host tissues or the pecan cultivation region. The grouping of selected isolates was confirmed by DNA sequence analysis. The *Alt a1* phylogeny corroborated no speciation within the dendrogram groups and showed 98–100% bootstrap similarity. This study reports the first documented rapid and reliable technique for routine screening identification of pathogens causing Alternaria black spot in South Africa.

## 1. Introduction 

*Alternaria* species are known plant pathogens that damage various plants, which in turn results in serious economic and agricultural losses. In particular, *Alternaria alternata* belongs to *Alternaria* sect. *Alternaria* [1], one of the most predominant species of the genus *Alternaria* [2,3]. The fungus produces colonies of dark mycelium with fast growth and a grey to olive or olive-brown colouration [4]. The surface texture of mature colonies appears fuzzy or woolly due to the presence of many hyphae [5]. *A. alternata* develops and grows as elongated chains of brown conidiophores that produce conidia [6]. Conidia are obclavate shaped (45–50 × 36 µm in size), with either single or long branched chains [1,7]. The conidia separate from the conidiophore and require dry, warm, windy weather (abiotic activity) or biotic activity to spread across the environment [8]. 

*A. alternata* is ubiquitous in nature and widely distributed [3,9]. It causes economically significant diseases on many crops. For instance, the fungus is known to cause black mould disease on tomatoes and citrus fruits in China [10,11]. *A. alternata* has become an emerging pathogen of oats (*Avena sativa*), causing spikelet rot disease, thereby affecting the grain yield and quality [12]. It has contributed to low blueberry production (*Vaccinium corymbosum*) in the United States of America (USA), causing postharvest disease known as blueberry fruit rot [13]. Muskmelon (*Cucumis melo*) is an important crop in Pakistan and is known to be attacked by *A. alternata*, which causes leaf spot disease [14]. *A. alternata* has been reported as the causative agent for leaf blight disease of little millet (*Panicum sumatrense*), a prominent cereal crop grown in India [15]. In South Africa, *A. alternata* is known to cause leaf blight on important crops such as sunflowers (*Helianthus annuus*) [16] and potatoes (*Solanum tuberosum*) [17]. 

*A. alternata* poses a threat to the South African pecan (*Carya illinoinensis*) industry [18]. The agricultural industry growing pecans has become popular because the nuts are quite valuable [19,20]. South Africa is the third-largest production country globally and exports over 25,000 tons annually [21]. However, the industry faces limited production of pecan nuts, primarily because of the development of diseases such as Alternaria black spot [18]. Usually, *A. alternata* occurs in pecans without causing any disease symptoms but becomes aggressive and causes disease during plant stress [22,23]. 

DNA analysis methods, especially PCR-based methods [24], represent a key step in accelerating the identification of *A. alternata* [1,3,25]. It is often difficult to identify species, especially those in *Alternaria*, because of the overlapping features. DNA sequence phylogenetic studies of the Alternaria major allergen (*Alt a1*) gene region were used for the characterisation and identification of the fungus [1]. The gene alone was not sufficient to stringently identify *Alternaria* species, especially because cryptic diverse species were described within the genus [26]. The internal transcribed spacer (*ITS*) and all other identified genes were insufficient to demonstrate strong species-groups of *Alternaria* [27,28]. Nine gene regions, namely SSU, *LSU*, *ITS*, *Gapdh*, *Rpb2*, *Tef1*, *Alt a1*, *EndoPG*, and *OPA10-2*, were used to revise the section *Alternaria* [3]. Many species were placed synonymous with *A. alternata sensu stricto* and grouped in one species complex in the section *Alternaria*. 

The application of DNA fingerprinting methods such as polymerase chain reaction–restriction fragment length polymorphism (PCR-RFLP) can be used to screen large collections of isolates. This application requires the combination of PCR and RFLP techniques to establish a method that is rapid, low-cost, high resolution, and convenient to use [29,30,31]. This method was widely studied for the identification of various fungal species [32,33]. For instance, the application has been used to analyse the genetic variation and diversity of *Fusarium oxysporum* f. sp. *fragariae*, known to cause Fusarium wilt of strawberries (*Fragaria ananassa*) [34]; *Paecilomyces variotii*, which causes dieback of pistachio (*Pistacia vera*) [35]; and *Alternaria* spp., which causes foliar diseases on potatoes (*S. tuberosum* L.) [36]. 

Multigene DNA sequence phylogeny was used to identify *Alternaria* isolates sampled from South African pecans with black spot symptoms as *A. alternata sensu stricto* [18]. No study has reported the complete geographic distribution of *A. alternata*, causing black spot disease on South African pecans. Accurately identifying such a large collection of isolates using DNA sequencing, especially when four genes must be used, is costly and time consuming. In this study, we developed and evaluated a PCR-RFLP assay to screen a large collection of *A. alternata* isolates (222) retrieved from different pecan localities in South Africa. The *Alt a1* gene region, previously found to be the most variable for *A. alternata* [18], and two restriction endonucleases (*Hae*III and *Hinf*I) were used to construct a genetic profile of the pathogen. The approach is useful to differentiate novel isolates into haplotypes, allowing an indication of variation. It also allows rapid and early detection of novel haplotypes which may also represent other *Alternaria* species. The data are then used to compare whether there is any significance between non-symptomatic and symptomatic pecan tissues. 

## 2. Material and Methods 

### 2.1. Sampling and Isolation of Alternaria 

Symptomatic (diseased) and non-symptomatic (apparently healthy) pecan plant organs (leaves, shoots, and nuts-in-shuck) showed symptoms consistent with black spot disease, such as circular black spot lesions and slight-deep sunken lesions, and all the isolates were dark brown to black with a white margin pigmentation [18]. These pecan tissues were collected from different commercial pecan orchards from eight provinces in South Africa during the 2017-to-2019 seasons (Figure 1 and Appendix A). Samples of the leaves, nuts, and twigs were transported to the laboratory following a protocol [18]. Plant material was surface-sterilised by washing with tap water, disinfected in 2% sodium hypochlorite (NaOCl) for 3–5 min, rinsed with sterilised distilled water (dH_2_O) for 2 min and then dried using laboratory tissue paper. Pure *Alternaria* cultures were obtained as previously described by Achilonu et al. [18]. 

### 2.2. DNA Extraction and PCR Amplification 

Genomic DNA (gDNA) was extracted from fresh mycelia (100 mg) of the *Alternaria* cultures (Table 1) using a ZR Quick-DNA Fungal/Bacterial MicropPrep™Kit (Zymo Research, Tustin, CA, USA). The DNA concentration and purity were determined with a NanoDrop Lite ND-2000 spectrophotometer (Thermo Fisher Scientific, Waltham, MA, USA). The DNA was standardised to a final concentration of 10 ng/µL for polymerase chain reaction. 

Polymerase chain reaction (PCR) was performed on the extracted gDNA using Alternaria major allergen (*Alt a1*) gene primers (Alt-For: 5′-ATG CAG TTC ACC ACC ATC GC-3′ and Alt-Rev: 5′-ACG AGG GTG AYG TAG GCG TC-3′) [37]. The PCR reaction (50 μL) consisted of 50–100 ng of template DNA, 0.3 μM of each primer, 2.5 mM MgCl_2_, 0.3 mM of each dNTP, and 1 U KAPA HiFi HotStart DNA Polymerase (Kapa Biosystems-Roche, Basel, Switzerland).

The T100TM Thermal Cycler conditions (Bio-Rad, Hercules, CA, USA) for PCR amplification entailed an initial denaturation step of 3 min at 95 °C followed by 24 cycles of 20 s at 98 °C, 30 s annealing step at 63.3 °C, 60 s at 72 °C and a final elongation step of 3 min at 72 °C. The PCR products stained with GelRed nucleic acid stain (Thermo Fisher Scientific) were examined using 2% agarose gel electrophoresis, visualised under a UV light “Gel Doc EZ Gel Documentation System” (Bio-Rad). 

### 2.3. Alt a1 PCR-RFLP Analyses 

For PCR-RFLP analysis, 10 µL of the amplified fragments were digested for 2 h with the restriction endonucleases *Hae*III and *Hinf*I, following the conditions recommended by the manufacturer (Thermo Scientific). Restriction fragments were analysed in 3% agarose gels stained with GelRed nucleic acid (Thermo Scientific) and visualised under a UV light “Gel Doc EZ Gel Documentation System” (Bio-Rad). 

For each fragment pattern, isolates were coded as ‘0’ or ‘1’, where ‘0’ meant absence and ‘1’ represented the presence of a band [34]. The unweighted pair group method with arithmetic average (UPGMA) and distance matrix (Euclidean) method dendrograms were determined through genetic variance between isolates using R-Studio v. 1.3.959 [38]. The agglomerative hierarchical clustering (compete) analysis and construction based on a linear combination of the variables were used to determine the maximum separation between isolates. Furthermore, the percentile data analyses of the isolates sampled from symptomatic and non-symptomatic pecan tissues were compared using Microsoft^®^ Excel 2020 v. 16.39.

### 2.4. Principal Coordinate Analyses 

The principal coordinate analysis (PCoA) between eight *A. alternata* populations was conducted based on the population genetic distance (GD) matrix for the haploid distance using GenAlEx v6.5 [39]. The analysis provided an alternative clustering approach to support the results of our UPGMA analyses. 

### 2.5. DNA Sequence and Phylogenetic Analysis 

Forty-two *A. alternata* isolates were selected from the six groups identified from the hierarchical clustering dendrogram to confirm their identities and the variation between the groups allocated by PCR-RFLP. The purified PCR amplicons for all the genes were sequenced in both primer directions using the BigDye Terminator v.3.1 kit (Applied Biosystems, Carlsbad, CA, USA). Reactions contained 2.5× BigDye Terminator Premix, 3.2 µM PCR primer, 5× sequencing buffer, and 3 µL PCR amplicon, making up a total volume of 10 µL. The conditions for the thermal cycler consisted of an initial denaturation step at 96 °C for 1 min, followed by 25 cycles of 96 °C for 10 s, 50 °C for 5 s, and 60 °C for 4 min. Cycle sequencing products were purified using the ZR DNA Sequencing Clean-Up^TM^ Kit (Zymo Research). The purified sequencing products were analysed with an ABI 3500xl Genetic Analyzer (Applied Biosystems), using standard protocols. The generated *A. alternata* DNA sequences (Appendix A) were aligned with the *Alt a1* DNA sequence dataset of [3] using MAFFT v. 7 [40]. A maximum likelihood (ML) phylogenetic tree with 1000 bootstrap replicates was constructed with MEGA X. v.10.1 [41], with the appropriate model obtained by MEGA. 

## 3. Results 

### 3.1. A. alternata Sampled from Pecan Tissues

The total number of host tissues and the statistical differences between *A. alternata* isolated from non-symptomatic and symptomatic pecan tissues are presented in Table 1. Eighteen fungal isolates (8.1%) were obtained from symptomatic nuts, Gauteng and North-West, with the least number of isolates (1) and most isolates (5) from the Northern Cape. The symptomatic leaves contained 133 isolates (59.9%), with the smallest number of isolates (1) from KwaZulu-Natal and the majority (49) from North-West. Seventeen *A. alternata* isolates (7.7%) originated from symptomatic twigs, a few isolates (1) were from Gauteng and North-West, and the majority (7) were from Limpopo. In total, 54 isolates (24.3%) were obtained from non-symptomatic leaves, and the least number of isolates (1) were from KwaZulu-Natal and the most isolates (38) were from North-West. 

### 3.2. Alt a1 PCR-RFLP Analyses 

The PCR products were approximately 980 bp in length and the *Hae*III and *Hinf*I digestion of the *Alt a1* gene amplicons showed distinct band profiles designated with letters for all 222 *A. alternata* isolates (Appendix A). All the isolates were cut by the restriction enzymes. The *Hae*III enzyme produced four different band patterns: A (600 bp), B (310 + 200 bp), C (410 bp), D (600 + 310 + 200 bp), with a fifth, pattern E (410 + 180 bp), produced by both enzymes, and another, pattern F (400 + 200 + 110 bp), produced by *Hinf*I (Figure 2). 

PCR-RFLP results of the combined restriction patterns of the two enzymes were corroborated by a hierarchical clustering dendrogram which showed the tree visualisation tool for all 222 *A. alternata* isolates (Figure 3). The analyses generated six groups that corresponded with the band patterns, and there was no association between groups, geographical location, or types of plant material. Group 1 had only three isolates from Gauteng (GP). Group 2 could be considered diverse because it had five isolates from Gauteng (GP), North-West (NW), Limpopo (LP), and Mpumalanga (MP). Group 3 and 4 contained 13 and 16 isolates from Limpopo, respectively, and showed no diversity. 

Group 5 was more diverse with 152 isolates from Limpopo (LP), KwaZulu-Natal (KZN), North-West (NW), Free State (FS), and Eastern Cape (EC). Group 6 consisted of 33 isolates from the Eastern Cape (EC) and Northern Cape (NC) and was also diverse. 

Only pecan cultivation regions showed a correlation grouping of the isolates when determining the correlation of genetic diversity. Specifically, 100% of the isolates from Gauteng and Limpopo were found only in group 1 and 4. *A. alternata* isolates from the other five regions were found in group 5 with a percentage comparison of 19% (Limpopo), 59% (North-West), 6% (Free State), 4% (KwaZulu-Natal), and 12% (Eastern Cape). 

### 3.3. Principal Coordinate Analyses 

The PCoA analyses among the eight populations of *A. alternata* isolates (Figure 4 A,B) depicted a cumulative based on covariance matrix with data standardisation for the haploid distance using GenAlEx. The isolates had a mixed distribution and were not clustered based on their geographic locations (Figure 4B). The majority of isolates from NC and KZN, respectively, were clustered within the third and fourth quadrant of the first PCoA axis. The second PCoA axis revealed few isolates from Limpopo (LP), and more isolates from North-West (NW) were spread across the second quadrant; the GP, EC, MP, and FS isolates were clustered across the first quadrant. The PCoA results agree with that of the cluster analysis (Figure 3) by also producing six distinct groups, but some isolates, substrates, and locations in PCoA were present in different groups. Therefore, there was no relationship between the *A. alternata* isolates, host tissue, and location. 

### 3.4. DNA Sequence and Phylogenetic Analysis 

The ML phylogenetic analysis generated a phylogenetic tree with 203 in-group and 1 out-group taxa. The 42 *A. alternata* isolates consistently grouped with other *A. alternata sensu stricto* sequences, forming part of the *A. alternata* species complex [3,18] (Figure 5). The results confirmed that the isolates of *A. alternata* sequences were clustered into six groups, with a bootstrap value of 98–100% within the groups. Phylogenetic analysis showed no association between clades, geographic location, or types of pecan tissues. 

## 4. Discussion 

This study established the identification of all isolates as *A. alternata* to determine the biogeographical distribution and degree of black spot disease of pecans in South Africa. The PCR-RFLP assay provided a rapid, cost-effective, and efficient diagnostic method. The overall cost of identifying *A. alternata* using PCR-RFLP analysis was calculated at 20–35 South African Rand (ZAR) per sample. Identification using PCR sequencing cost ZAR 150 per sample. Furthermore, the PCR-RFLP analysis proposed in the study is not only cheaper but also a much faster diagnostic option, with a turnaround time of less than 3 h, while sequencing, as an outsourced service, takes at least 3 days to complete. Thus, compared to single- or multi-locus sequencing, PCR RFLP-based typing is technically less demanding, takes less time, and has low costs when assessing the variability in the isolated *A. alternata* genotypes, and confirmed to be all *A. alternata*. The dendrogram and PCoA analyses showed six clusters of these isolates. The maximum likelihood phylogenetic tree revealed the same number of clades, and thus, the isolates were grouped with the *A. alternata* species complex. Therefore, the findings confirmed that the *A. alternata* isolates how a consistently prevalent genetic variation within their population. 

The combination of two endonucleases through dendrogram analysis of *Alt a1*-RFLP profiles enabled the identification of these 222 *A. alternata* isolates into six groups or clusters. The banding patterns within the *Alt a1* gene region of the 222 *A. alternata* isolates after digestion with the *Hae*III and *Hinf*I endonuclease enzymes showed different restricted-fragment ranges. The combination of the two restriction enzymes was sufficient to discriminate between the isolates and reproduce better banding composite profiles [36], and it has been used to positively identify *A. alternata* isolates from grapes [42], pistachio [43], and apple [44]. 

The genetic variation in the *A. alternata* isolates from the eight geographical locations showed low polymorphism within the *A. alternata* isolates, which could have been a result of episodes of evolutionary processes such as mutation, recombination, and migration [26,45]. Most isolates from the North-West, Free State, Kwazulu-Natal, Eastern Cape, and Northern Cape provinces were only found in group 5, showing correlations between their genetic similarity. The results for PCoA analyses supported the evidence of a lack of genetic structure of *A. alternata* isolates within the eight populations. At this point, there was no comparison based on pathogens and the different host types because of the inconsistent isolation of *A. alternata* from the pecan organs. These findings are consistent with human-mediated gene flow occurring across pecan production locations that share genetic information over time and are likely to show greater genetic similarity [46]. 

We observed a similarity in *A. alternata* isolate clusters despite geographical distances of 405 km to 993 km between the eight pecan cultivation regions (https://www.google.com/maps/dir/, accessed on 20 November 2020). This could be because the long-distance isolation observed throughout the entire study makes it difficult for the fungal spores to be dispersed by wind. The grouping of these isolates with a distant region could have been due to the transfer of the fungus from pecan nurseries to pecan orchards, or from pecan orchards to other orchards. This is because of the tendency of pecan producers to purchase grafted trees from nurseries within and between different regions as pecan production and cultivation has since increased in Mpumalanga, Limpopo, North-West, and Northern Cape provinces to the Eastern Cape, Free State, and Gauteng provinces over the years. Moreover, pecan nurseries still use self-seeding plants and grafting for cultivation [47], from where *A. alternata* can originate. Therefore, the present study suggests that human-mediated dispersal may have played an important role in the dynamics of the distribution of *A. alternata*. 

The symptomatic and non-symptomatic leaf samples clearly showed distinct presences of *A. alternata* relative to the symptomatic nut and twig samples. The abundance of fungal isolates was significantly higher in the leaf organ than in the nut and twig organs. Unlike nuts and twigs, the light texture of pecan leaves could influence the contact and permeability of spores to cause black spot disease, especially after secondary damage by insects, humans, or unfavourable climatic conditions [48,49]. This could be the cause of fungi being dependent on tissue-specific interactions [50,51], though we did not investigate the successions of *A. alternata* on various tissues of pecan, which deserves further study. 

*Alt a1* sequence alignments and the ML phylogenetic tree were used to assess genetic differentiation within individuals of the *A. alternata* isolates. Discrimination based on the high sequence diversity of the *Alt a1* gene was observed in the *A. alternata* isolates indicating genetic homogeneity [37,52]. A similar deduction was reached with genetic differentiation among similar morphological types of *A. alternata* based on *Alt a1* gene encoding isoenzyme analysis [1,8,53]. No speciation within the fungal isolates was confirmed from their respective phylogenetic tree clades, suggesting that *A. alternata* is the most commonly occurring species found across pecan plantations in South Africa. Generally, *A. alternata* reproduces by propagating through asexual spores; thereby, the fungus becomes less diverse because of the presence of a single mating type (or MAT) locus [54,55,56]. 

## 5. Conclusions 

The proposed diagnostic capabilities of the PCR-RFLP method were successful in addressing our knowledge gaps about the fungus *A. alternata*, as it is fast, efficient, and a cheaper tool for advanced screening identification of *A. alternata* isolates than DNA sequencing and can be applied to a large number of fungal isolates in a short time. Owing to the ubiquitous nature and ability of *A. alternata* to cause disease such as black spot in various plants, the fungus has become increasingly problematic in agriculture. Disease management of black spot disease on pecans in South Africa should include the identification of disease threats and an understanding of the pathotypic diversity and level of *A. alternata* virulence. The results of this work are intended to provide a benchmark for future population studies and estimates of the genetic diversity and distribution in the *A. alternata* population across main pecan production areas in South Africa.

## Figures and Tables

**Figure 1 genes-14-01115-f001:**
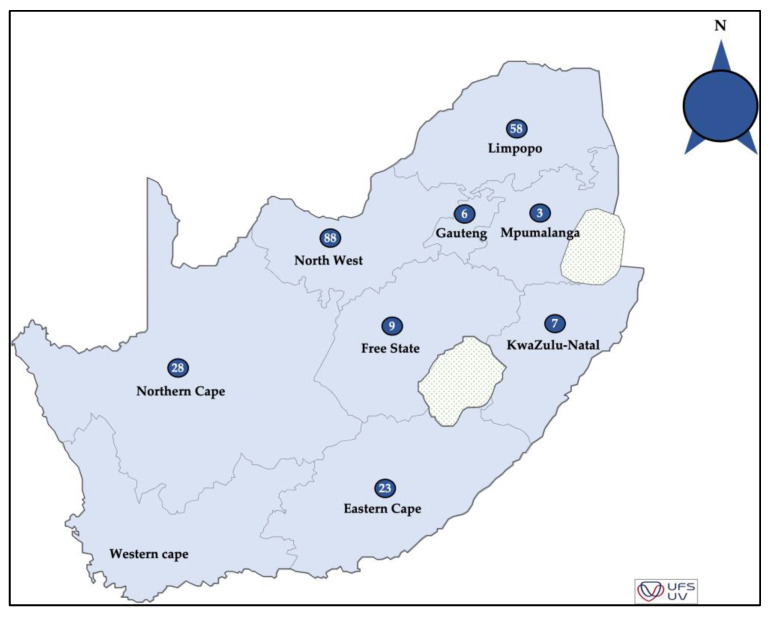
The distribution map showing the locations of pecan orchards and the number of *A. alternata* isolates (blue circles) sampled in eight provinces of South Africa.

**Figure 2 genes-14-01115-f002:**
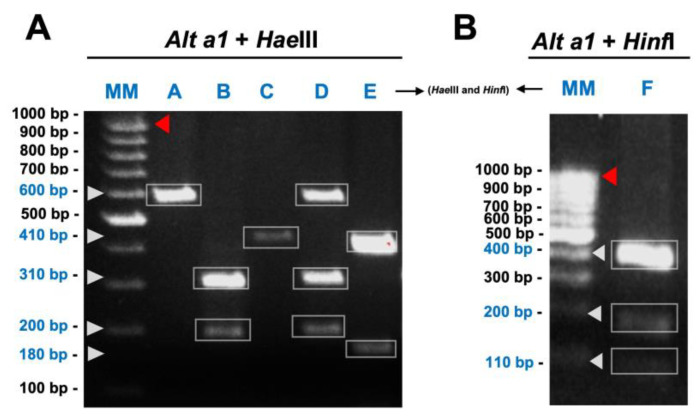
Restriction fragment length polymorphism banding patterns. (**A**) 3128 Fragmentation of Alternaria major allergen (*Alt a1*) gene with *Hae*III, with banding 3129 profile: (A) 600 bp, (B) 310 bp + 200 bp, (C) 410 bp, (D) 600 bp + 310 bp + 200 bp, 3130 (E) 410 bp + 180 bp and (E) 410 bp + 180 bp. (**B**) Fragmentation of Alternaria major allergen (*Alt a*1) gene with *Hinf*I, with banding profile: (F) 400 bp + 200 bp + 110 bp. 3132 Lane MM = Molecular marker (100 bp DNA ladder) (Thermo Fisher Scientific, Waltham, MA, USA). Uncut band size (PCR amplicon: 980 bp) indicated with red arrow. Restriction fragment band size (Blue colour MM (bp)) indicated with white arrow.

**Figure 3 genes-14-01115-f003:**
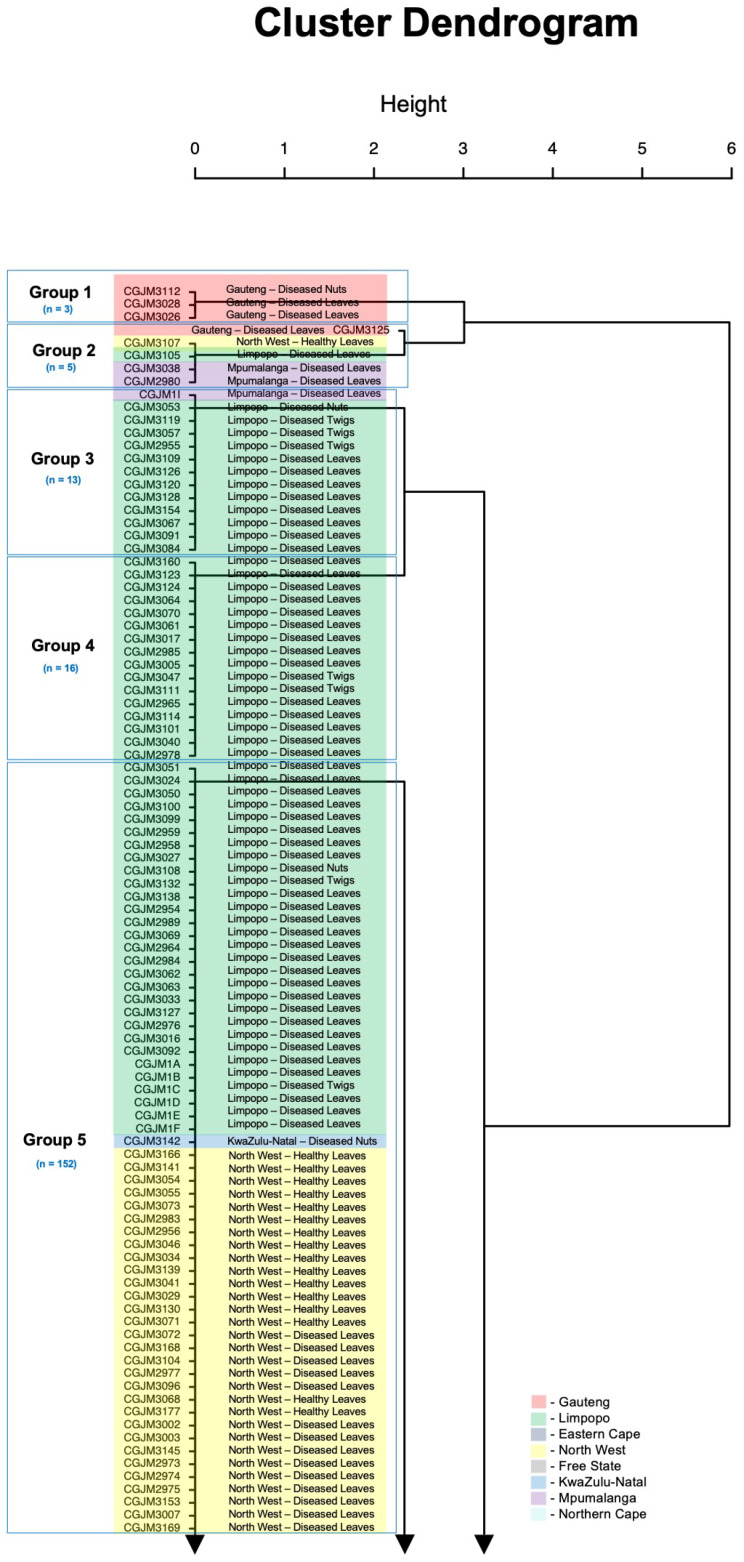
Hierarchical clustering dendrogram using “complete” agglomeration method, obtained based on Euclidean distance matrix. The asterisk (*) represents data matrix. The y-axis represents the Euclidean distance (rescaled distance cluster combination) between the *A. alternata* isolates. The different groups and colours depict the six clusters and “n” represent total number of isolates in a group.

**Figure 4 genes-14-01115-f004:**
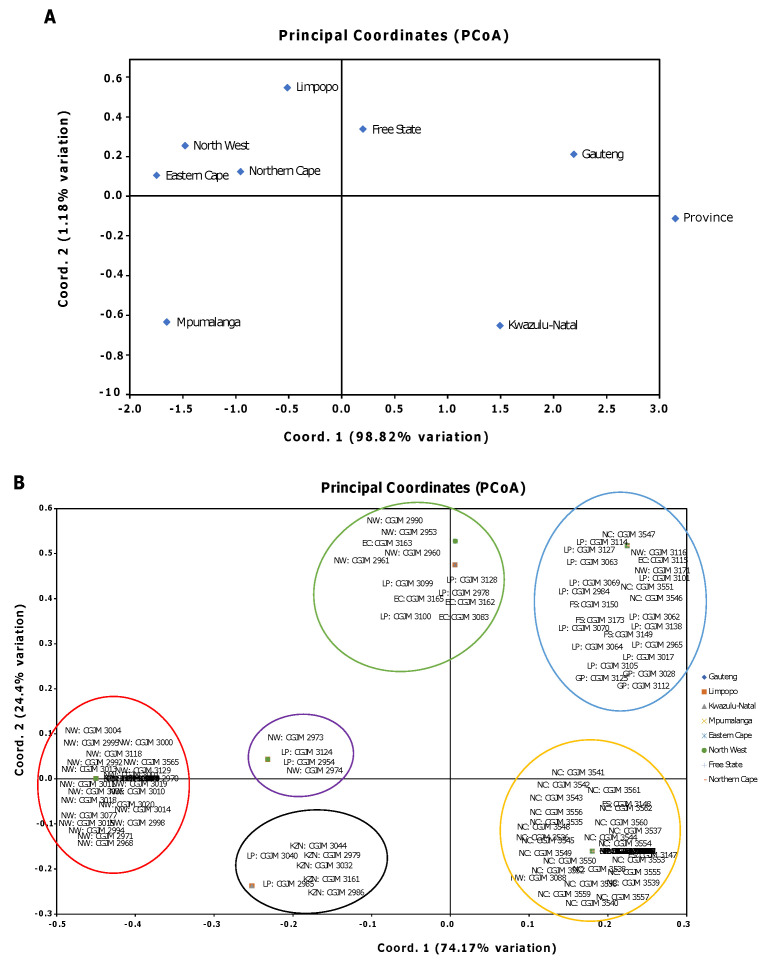
Principal coordinates analysis (PCoA) among eight populations of *A. alternata* based on the mean population genetic distance matrix for haploid distance using GenAlEx. Different colours represents the six clusters. (**A**) The diamond symbols indicate the eight populations from Mpumalanga (MP) and KwaZulu-Natal (KZN), respectively, clustered within the 3rd and 4th quadrant of the first PCoA axis. The second PCoA axis contained isolates from Eastern Cape (EC), Limpopo (LP), Northern Cape (NC) and North-West (NW) spread across the 2nd quadrant and then Gauteng (GP) and Free State (FS) isolates were clustered across the 1st quadrant. (**B**) *A. alternata* isolates had a mixed distribution into six clusters and were not confined to their locations.

**Figure 5 genes-14-01115-f005:**
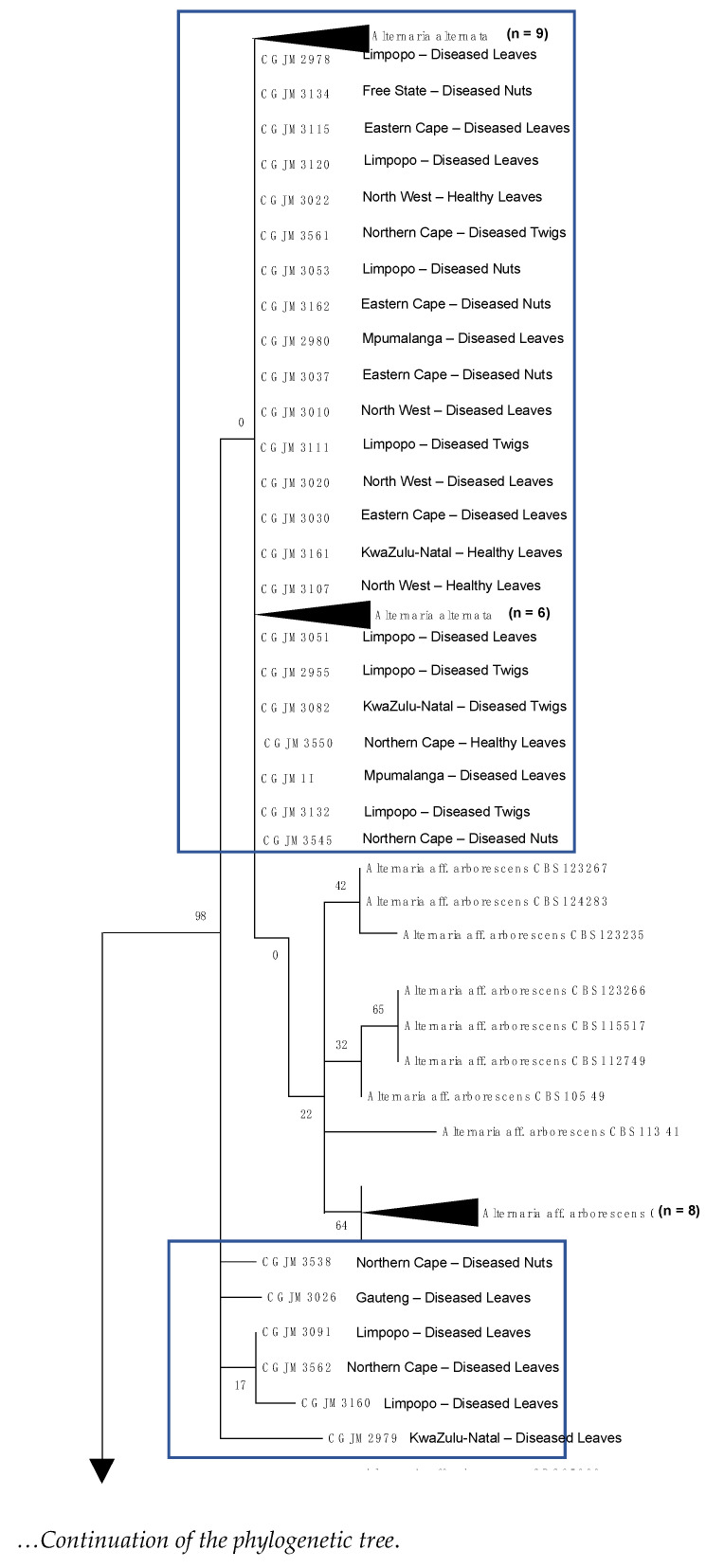
Principal phylogenetic tree based on the *Alt a1* gene sequence of *A. alternata* isolates generated by maximum likelihood. Bootstrap support values are indicated on the branch nodes below branches. The 42 identified *A. alternata* isolates are highlighted in grey frames; provinces of origin and different plant organs are indicated. The black arrow indicates the total number (n) of the same fungal species.

**Table 1 genes-14-01115-t001:** Number of *A. alternata* isolates from symptomatic leaves, nuts-in-shuck, shoots and non-symptomatic leaves.

Location	Symptomatic Nuts-in-Shuck	Symptomatic Leaves	Symptomatic Shoots	Non-Symptomatic Leaves
Gauteng	1	4	1	0
Limpopo	2	48	7	0
Kwazulu-Natal	3	1	2	1
Mpumalanga	0	3	0	0
Eastern Cape	3	14	2	4
North West	1	49	1	38
Free State	3	3	0	3
Northern Cape	5	11	4	8
Total *	18	133	17	54
Percentage	8.1%	59.9%	7.7%	24.3%

Total *: number and percentage of *A. alternata* isolated from symptomatic or non-symptomatic pecan organs.

## Data Availability

All data generated or analysed during this study are included in this published article as Appendix A.

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
