# Peer review of "Differential Detection of *Alternaria alternata* Haplotypes Isolated from *Carya illinoinensis* Using PCR-RFLP Analysis of *Alt a1* Gene Region"

_genes, 2023, doi:10.3390/genes14051115_

Round 1

Reviewer 1 Report

Top of Form

The causative agent of  ‘aTop of Form

ternaria black spot disease in pecan’ is Alternaria alternata, a pathogen that may cause serious danger to pecan industry around the world. A number of molecular diagnostic applications are available for fungal diseases analysis. In current manuscript author have presented application of DNA polymorphism from A. alternata isolates from S. Africa region. Authors used PCR-based RFLP genetic characterization of specific DNA loci in pathogen (coupled with restriction endonucleases). The results show convincing evidence regarding diversity of A. alternate independent of pecan populations and geographical restrictions.

Authors provides documented proofs for rapid and reliable screening of pathogens causing Alternaria black spot in South Africa. Bottom of Form

Top of Form

The study presents a timely report on pecan pathogen’s diversity, its understanding and it is significant for efficient control of disease programs. The study could be considered for publication after careful revisions.

Some comments:

1.       Figure 2 and 4 authors could reorganize and could provide a dendrogram based location -1 sample per location. Because each location may not show that much diversity.

If it is difficult to sort out, then keep as it is.

2.       More gel picture could shown and region wise diversity in bands could be displayed.

3.       Authors need to update literature on this topicof molecular markers in plant breeding

4.       For example., : Multiplex molecular marker-assisted analysis of significant pathogens of cotton (Gossypium sp.), 2023; Biocatalysis and Agricultural Biotechnology; Microsatellite and RAPD analysis of grape (Vitis spp.) accessions and identification of duplicates/misnomers in germplasm collection, Upadhyay et al., 2010 Indian J Hortic Volume 67 Pages 8-15; Microsatellite analysis to differentiate clones of Thompson seedless grapevine, Upadhyay et al., 2010, Ind Journal of Horticulture, Volume 67 Issue 2 Pages 260-263. Chavhan, R.L., Hinge, V.R., Kadam, U.S. et al. Real-time PCR assay for rapid, efficient and accurate detection of Paramyrothecium roridum a leaf spot pathogen of Gossypium species. J. Plant Biochem. Biotechnol. 27, 199–207 (2018).

5.       If available, phenotypic data or morphological features of pathogen and disease status in pecan could shown as figures or pictures.

6.       Physical geographical location of selected regions could be displayed on a map, insert map in figure 1 or 2.

Language and content is fine.

Author Response

Reviewer #1

  • Figure 2 and 4 authors could reorganize and could provide a dendrogram based location -1 sample per location. Because each location may not show that much diversity. If it is difficult to sort out, then keep as it is.

Response:

Thank you for the comments. The purpose of Figure 4 was to corroborate by DNA sequence phylogeny that the clustered isolates from the dendrogram (Figure 3) is A. alternata.

  • More gel picture could shown and region wise diversity in bands could be displayed.

Response:

Thank you. Due to the limited space in the manuscript, we had to show the representative restriction enzyme digestion band patterns and then place all band patterns for the isolates in the Supplementary Materials (Figure S1).

  • Authors need to update literature on this topic of molecular markers in plant breeding.

Response:

Thank you. All suggested articles have been included in the manuscript,

Line 73 – Reference: [24] and Line 87- Reference: [30 & 31].

For example.:

  • Chavhan, R. L., Hinge, V. R., Kadam, U. S., Kalbande, B. B., and Chakrabarty, P. K. 2018. Real-time PCR assay for rapid, efficient and accurate detection of Paramyrothecium roridum a leaf spot pathogen of Gossypium J Plant Biochem Biotechnol. 27:199–207.
  • Chavhan, R. L., Sable, S., Narwade, A. V., Hinge, V. R., Kalbande, B. B., Mukherjee, A. K., et al. 2023. Multiplex molecular marker-assisted analysis of significant pathogens of cotton (Gossypium). Biocatal Agric Biotechnol. 47:102557.
  • Upadhyay, A., Kadam, U. S., Chacko, P. M., Aher, L., and Karibasappa, G. 2010. Microsatellite analysis to differentiate clones of Thompson Seedless grapevine. Indian Journal of Horticulture. 67:260–263.

  • If available, phenotypic data or morphological features of pathogen and disease status in pecan could shown as figures or pictures.

Response:

Thank you for the suggestion. We have a recent published article that depicted the phenotypic and morphological features of the pathogen.

Lines 66-67 and Lines 69-71, Reference: [18].

  • Achilonu, C.C.; Marais, G.J.; Ghosh, S.; Gryzenhout, M. Multigene Phylogeny and Pathogenicity Trials Revealed Alternaria Alternata as the Causal Agent of Black Spot Disease and Seedling Wilt of Pecan (Carya Illinoinensis) in South Africa. Pathogens 2023, 12, 672–691, doi:10.3390/PATHOGENS12050672.

  • Physical geographical location of selected regions could be displayed on a map, insert map in figure 1 or 2.

Response:

Thank you for the suggestion. We have inserted a distribution map as Figure 1.

Reviewer 2 Report

A methodologically sound paper convincingly documents the possibility of PCR-RFLP application in polymorphisms detection among Alternaria alternata isolates and identification this species within Alternaria population occupying Carya illinoinensis. The authors should clarify two aspects:

The authors explain variation among A.alternata isolates by recombination (L 281 – L 283). What do the authors mean? The main sources of  genetic recombination are  independent assortment of alleles and  crossing over, which took place during meiosis, unfortunately occurrence of Alternaria alternatata teleomorph is not documented.

 The Authors wrote (L. 325), that “The proposed diagnostic capabilities of the PCR-RFLP method was successful in addressing our knowledge gaps about the fungus A. alternata, as it is fast, efficient, and a cheaper tool for advanced screening identification of A. alternata isolates than the DNA sequencing, and can be applied to a large number of fungal isolates in a short time”

According to the supervisor, this conclusion is too general. The Alt a1 gene region is present in several species that do not belong to A.alternata, i.e. the complex of species Alternaria arborescens, Alternaria tomato, Alternaria jacinthicola, Alternaria eichhorniae, Alternaria betae-kenyensis. These species were not included in the analysis. How do authors know that the proposed method will distinguish them from A.alternata

Author Response

Reviewer #2

The authors should clarify two aspects:

  • The authors explain variation among alternata isolates by recombination (L 281 – L 283). What do the authors mean? The main sources of genetic recombination are independent assortment of alleles and crossing over, which took place during meiosis, unfortunately occurrence of Alternaria alternata teleomorph is not documented.

Response:

Thank you for the comment in now lines 391-394. “The genetic variation of the A. alternata isolates from the eight geographical locations showed low polymorphism within the A. alternata isolates, which could have been as a result of episodes of evolutionary processes like mutation, recombination, and migration”.

Responding to your question, almost all fungi, including A. alternata, have the ability to reproduce asexually through the formation of mitotic spore stages, or to reproduce sexually through a process of meiotic recombination to produce meiotic spores. See updated references [46 & 47].

  • Eschenbrenner, C.J.; Feurtey, A.; Stukenbrock, E.H. Population Genomics of Fungal Plant Pathogens and the Analyses of Rapidly Evolving Genome Compartments. Methods in Molecular Biology 2020, 2090, 337–355, doi:10.1007/978-1-0716-0199-0_14/FIGURES/1. 47.
  • Meng, J.W.; Zhu, W.; He, M.H.; Wu, E.J.; Duan, G.H.; Xie, Y.K.; Jin, Y.J.; Yang, L.N.; Shang, L.P.; Zhan, J. Population Genetic Analysis Reveals Cryptic Sex in the Phytopathogenic Fungus Alternaria Alternata. Scientific 2015, 5, 1–10, doi:10.1038/srep18250.
  •  
  • The Authors wrote (L. 325), that “The proposed diagnostic capabilities of the PCR-RFLP method was successful in addressing our knowledge gaps about the fungus alternata, as it is fast, efficient, and a cheaper tool for advanced screening identification of A. alternata isolates than the DNA sequencing, and can be applied to a large number of fungal isolates in a short time”

According to the supervisor, this conclusion is too general. The Alt a1 gene region is present in several species that do not belong to A. alternata, i.e. the complex of species Alternaria arborescens, Alternaria tomato, Alternaria jacinthicola, Alternaria eichhorniae, Alternaria betae-kenyensis. These species were not included in the analysis. How do authors know that the proposed method will distinguish them from A.alternata.

Response:

Thank you for the comments. Based on our study, we have only evaluated those A. alternata isolates associated with ABS on pecans in South Africa, see Lines 66-67 and Lines 69-71, Reference: [18], and we corroborated have our dendrogram clustered isolates (Figure 3) by DNA sequence phylogeny (Figure 5) using representative isolates and reference DNA sequences of A. arborescens, A. alstroemeria, A. jacinthicola, and A. burnsii.